# Study on the Unbalanced Curl Seal Failure of the Magnetorheological Fluid Sealing Device of the Hydraulic Turbine Main Shaft under Different Speed Abrupt Conditions

**Jie Cheng** **, Zheng-Gui Li \*, Yang Xu, Wang-Xu Li and Xin-Rui Li**

Key Laboratory of Fluid and Power Machinery of Ministry of Education, Xihua University,
Chengdu 610039, China; chengjie@stu.xhu.edu.cn (J.C.); xy@stu.xhu.edu.cn (Y.X.);
liwangxu@stu.xhu.edu.cn (W.-X.L.); lixinrui@stu.xhu.edu.cn (X.-R.L.)
\* Correspondence: lzhgui@mail.xhu.edu.cn; Tel.: +86-176-0283-2345

**Abstract:** The fluid flow in the runner of a hydraulic turbine has serious uncertainties. The sealing failure of the magnetorheological (MR) fluid sealing device of the main shaft of the hydroturbine, caused by a sudden change in speed, has always been a difficult topic to research. This study first derives the MR fluid seal pressure and unbalanced curl equations of the hydroturbine main shaft, and then analyzes the seal pressure and friction heat under different rotational speed mutation conditions through experiments. After verification, the temperature field and magnetic field distribution of the MR fluid sealing device of the main shaft of the hydraulic turbine are obtained via numerical calculation. The results show that the external magnetic field affects the magnetic moment of the magnetic particles in the MR fluid, resulting in a significant change in frictional heat, thereby reducing the saturation of magnetic induction intensity of the MR fluid. This results in a decrease in the sealing ability of the device. The size and abrupt amplitude of the main shaft of the hydraulic turbine, and friction heat is positively correlated reducing the sealing ability of the device and causing sealing failure. Based on our results, we recommend adding the necessary cooling to the device to reduce the frictional heat, thereby increasing the seal life of the device.

**Keywords:** frictional heat; hydraulic turbine main shaft; MR fluid seal; different speed abrupt conditions; seal failure; unbalanced curl

## 1. Introduction

With the rapid development of the world economy, energy demand has rapidly increased. To realize the sustainable development of society and protection of the Earth, green and environmentally friendly renewable energy has become an important research topic. According to the World Energy Development Report (2020) report released by the China International Energy Security Research Center, the global installed hydropower capacity has reached 1.308 billion kW, ranking first in the world's list of renewable energy [1]. Regarding hydropower, the sealing performance of the main shaft of the core equipment turbine is directly related to the safety and reliability of the entire power generation process. Presently, the main shafts of hydraulic turbines mostly use rubber, packing, floating-ring, and labyrinth seals [2]. However, these sealing methods have several problems, including having complex structures and low reliability and being easily worn down. As a new type of sealing method, magnetorheological (MR) fluid sealing has become a research hotspot in the sealing industry, owing to its high reliability, low viscous friction, and good self-healing properties [3,4]. MR fluids are mainly comprised of micron-sized magnetic solid particles, surfactants, and base fluids. Compared with magnetic fluid seals, magnetic solid particles sealed by MR fluids are much larger. Rheological fluid sealing is mainly aimed at large-gap sealing [5,6], mainly employing the magnetic force of an external magnetic field to gather and align magnetic solid particles in the fluid into chains, to form a semi-solid having an anti-shearing effect [7,8].

Research on the sealing of MR fluids can be traced back to the 1960s. Rosensweig [9] conducted research on sealing the movable parts of a spacesuit in a vacuum environment, and applied it in practice. Thereafter, scholars worldwide have conducted extensive research on MR fluid sealing. For example, in 1979, Rosensweig [10] studied the dynamic characteristics of MR fluids, and found that its liquid phase has one more source stress than ordinary fluid mechanics. In 1987, Shul'man [11] studied the rheological properties and shape parameters of MR fluids in relation to energy dissipation and heat transfer of rotating magnetic fields. In 1996, Kordonski [12] compared magnetic and MR fluid seals, determining that the sealing ability of MR fluid seals is much higher than that of magnetic fluid seals, owing to shearing. In 2004, Iyengar [13] designed a method to test the wear of MR fluids. The process provided a theoretical reference for the future study of MR fluid seals. In 2012, Potoczny [14] studied the relationship between the volume of MR fluid and sealing pressure, obtaining expressions of MR fluid volume and sealing pressure. In 2013, Zhou [15] applied MR fluid seals to the steel industry, solving the air-leakage problem of circular coolers, and expanded the application of MR fluid seals. In 2017, Susan–Resiga [16] experimentally studied the apparent viscosity and magnetization of different magnetic induction intensities, shear rates, magnetic solid particle sizes, and volume fractions to obtain an infinite measure of the sealing performance of MR fluids. The characteristic parameters of the Mersenne (Mn) and Carson (Ca) numbers provide theoretical references for the practical application of MR fluid seals. In 2018, Zhang [17] again compared magnetic and MR fluid seals, finding that the MR fluid seal has higher pressure resistance. Additionally, the sealing pressure and sealing time of the MR fluid were given relevant expressions. In 2019, Kubík [18] designed a new type of shrinkable MR fluid sealing structure that has a low friction torque and high sealing pressure. In 2020, Yang [19] optimized the design of a four-magnetic-source MR fluid seal, and studied the sealing gap, ratio of the height and length of the permanent magnet, and height and width of the pole teeth. The influences of the structural parameters (e.g., groove width, and pole tooth width ratio) on sealing performance led to new research ideas for magnetic fluid sealing. In 2021, Wang [20] studied the O-rings of MR fluid dampers; the friction and wear mechanism was studied, and a series of friction force series were established. The relationship between piston-rod speed, particle size, particle mass fraction, friction coefficient, and film thickness was obtained through calculations and analysis. In 2021, Yuqing Li [21] selected MR fluid materials and used the proportioning method and stability principle to study MR fluid seals. By summarizing the performance methods, a formula for MR fluids with excellent performance was obtained, ultimately improving the performance of MR fluid seals.

Currently, the literature that has been consulted shows that there is no study on the seal failure caused by the unbalanced curl of the MR fluid seal device of the hydroturbine main shaft, owing to sudden changes in rotational speed. Therefore, this study first derives the MR fluid seal pressure and unbalanced curl equations of the turbine main shaft, then obtains the pressure values and frictional heat under different speed mutation conditions through experiments. Subsequently, we verify the experimental results and numerical calculation results. Finally, the magnetic field and temperature field distribution of the MR fluid sealing device are obtained using numerical calculation, which provides a new theoretical reference for the research on MR fluid sealing.

## 2. Theory

### 2.1. MR Fluid Seal Pressure Equation

This study does not consider the change in the viscosity of the MR fluid. The MR fluid sealing device for the main shaft of a hydraulic turbine is shown in Figure 1. It is mainly composed of magnetic poles, a main shaft, permanent magnets, insulating materials, and MR fluid. The magnetic energy of the permanent magnet emerges from the N level, and passes through the magnetic pole, MR fluid, and main shaft, and finally, through the other side of the magnetic pole to return to the permanent magnet S level. The MR fluid located

between the main shaft and magnetic poles (sealing gap) is subjected to magnetic force, and the magnetic particles in the MR fluid gather into chains to prevent leakage of the sealed medium and achieve the purpose of sealing.

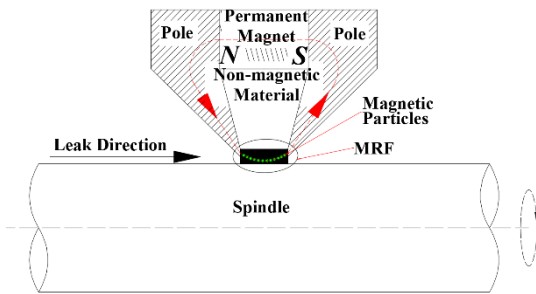

**Figure 1.** MR fluid sealing device.

The N-S equation of the MR fluid [22] is

$$\rho_m \frac{\partial V}{\partial t} + \rho_m (V \cdot \nabla) V = -\nabla p + \eta_H \nabla^2 V + \mu_0 M \nabla H, \tag{1}$$

where $\rho_m$ is the density of the MR fluid, V is the linear velocity of the MR fluid, $p$ is the magnetic fluid pressure, $\mu_0$ is the vacuum permeability, $\eta_H$ is the viscosity of the MR fluid in an external magnetic field, $M$ is the magnetization, and H is the strength of the external magnetic field.

When calculating the sealing pressure of the MR fluid, this study does not consider the internal degrees of freedom of the MR fluid, as the external magnetic field does not cause the solid particles to rotate. The MR fluid moves circularly around the central axis of the spindle in the gap. Therefore, this study uses a cylindrical coordinate system with the central axis of the spindle as the z-axis. We thus expand (1) to obtain

$$\begin{cases} \frac{\partial p}{\partial r} = \rho_m \frac{v_\theta^2}{r} + \frac{\partial}{\partial r} \mu_0 \int_0^H M dH \\ 0 = \frac{\partial v_\theta^2}{\partial r^2} + \frac{1}{r} \frac{\partial v_\theta}{\partial r} + \frac{\partial^2 v_\theta}{\partial z^2} - \frac{v_\theta}{r^2} \\ 0 = -\frac{\partial p}{\partial z} + \frac{\partial}{\partial z} \mu_0 \int_0^H M dH \end{cases} . \tag{2}$$

As shown in Figure 1, assuming that the spindle speed is $\omega$, the spindle radius is $R$, and the sealing gap is $Lg$. When the rotation radius $r = R$, then V = $R\omega$; when the rotation radius $r = R + Lg$, then V = 0. Substituting the boundary conditions into (2) obtains

$$v_\theta = \omega_0 \left( C_1 r + \frac{C_2}{r} \right), \tag{3}$$

where $r$ is the radius of a point in the MR fluid, $C_1 = -\frac{R^2}{L_g(2R+L_g)}$, and $C_2 = \frac{R^2(R+L_g)^2}{L_g(2R+L_g)}$.

By substituting (3) into (2) to fully integrate the pressure term, we obtain

$$p = \mu_0 \int_0^H M dH + \phi(r) + C, \tag{4}$$

where $\phi(r) = \frac{1}{2} \rho_m \omega^2 \left( C_1^2 r^2 - \frac{C_2^2}{r^2} + 4 C_1 C_2 \ln r \right)$.

For MR fluid seals, the yield stress of the MR fluid also has a certain resistance to pressure. Therefore, we first obtain the MR fluid pressure difference according to (4), and then superimpose the MR fluid yield stress [23]. Finally, the total pressure of the MR fluid seal can be obtained as

$$\Delta p = M_S(B_A - B_B) + \phi(r_A) - \phi(r_B) + 3.89 n D R_0^2 (1 - \varepsilon)^{0.75} M^2 \sin \alpha \frac{3L}{b}, \tag{5}$$

where $M_S$ is the saturation magnetic induction intensity of the magnetic fluid, $B_A$ is the magnetic induction intensity on the high-voltage side, $B_B$ is the magnetic induction intensity on the low-voltage side, $r_A$ is the radius of the high-pressure side, $r_B$ is the radius of the low-pressure side, $D$ is the demagnetization coefficient of magnetic particles, $R_0$ is the radius of the magnetic particle, $\varepsilon$ is the porosity, $\alpha$ is the maximum deflection angle of the flux linkage under the action of an external magnetic field, L is the axial length of the gap filled with MR fluid, and b is the radial length of the gap filled with MR fluid.

### 2.2. MR Fluid Friction Power Loss Equation

Figure 2 shows a schematic of the shear flow of the MR fluid. Figure 2a shows the shear flow of the MR fluid under the condition of an external magnetic force H = 0. Figure 2b shows the MR fluid under the condition of an external magnetic force H ≠ 0. When the external magnetic force H = 0, the MR fluid moves circularly around the z-axis under the action of the centrifugal force of the main shaft of the hydraulic turbine. When the external magnetic force H ≠ 0, the MR fluid moves under the action of the centrifugal force of the main shaft of the hydraulic turbine and the external magnetic force. An unbalanced curl, $\Omega \neq \frac{1}{2}(rot\ v)$, is produced, which increases the friction between the MR fluid and main shaft of the hydraulic turbine, forming higher frictional heat, thereby reducing the saturation magnetic induction intensity of the MR fluid as well as the sealing pressure, eventually leading to reduced sealing.

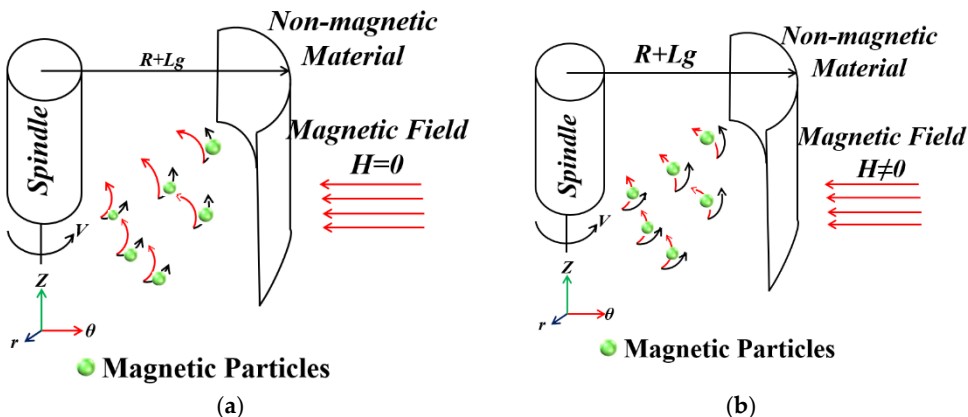

**Figure 2.** Shear motion of the MR fluid: (**a**) H = 0; (**b**) H ≠ 0.

From Figure 2a, given the shear motion of the MR fluid under the condition of external magnetic field H = 0, the motion equation of the MR fluid under this condition is

$$\mathbf{v}(r) = v\frac{z}{R + L_g}(\mathbf{e}_r + \mathbf{e}_\theta),\tag{6}$$

where $v$ is the rotational linear velocity of the main shaft of the turbine, $z$ is the z-axis coordinate, $\mathbf{e}_r$ is the basis vector in the $r$ direction, and $\mathbf{e}_\theta$ is the basis vector in the $\theta$ direction. Based on Figure 2b, for the shear motion of the MR fluid under the condition of an external magnetic force H ≠ 0, the curl, $\Omega$, under the action of the external magnetic force is

$$\Omega(\mathbf{r}) = \frac{1}{2}rot\ \mathbf{v} = \frac{v}{2(R + L_g)}\mathbf{e}_z,\tag{7}$$

where $e_z$ is the basis vector in the z-direction. By using (7), the rotational angular velocity, $\omega 0$, of the MR fluid under the action of an external magnetic field can be obtained as

$$\omega_0 = \frac{fM(H)\mu_0 H}{\mu_r + fM(H)\mu_0 H}\Omega(r)\mathbf{e}_\theta,\tag{8}$$

where $f$ is the relaxation time of magnetization, $\mu_r$ is the relative permeability of the magnetic fluid, $H$ is the intensity of the external magnetic field, and $M(H)$ is the magnetization intensity of the MR fluid.

For the MR fluid subjected to an external magnetic force, there will be an offset, $M$, in the $z$- and $r$-axis directions. The offset, $M$, can be expressed using the Shlionmis formula.

$$\mathrm{M} = \omega_0 M(H)(-\mathrm{e}_r + \mathrm{e}_z). \tag{9}$$

Under the action of an external magnetic field, the friction balance equation of the MR fluid seal is

$$\int dV_m \mu_0 \mathrm{M} \times \mathrm{H} = \int dV_m Z_r \omega_0, \tag{10}$$

where $V_m$ is the volume of the MR fluid, and $Z_r$ is the rotational viscosity of the magnetic fluid.

To facilitate the calculation, the function, $K(H)$, is introduced to obtain

$$K(H) = V_m \mu_0 H_0 M(H) K_0 \Delta T, \tag{11}$$

where $H_0$ is the magnetic-field strength in the MR fluid, $K_0$ is Boltzmann's constant, and $\Delta T$ is the extra torque generated by the MR fluid under the action of an external magnetic field.

According to the magnetization formula of MR fluid, the magnetization intensity [24], $M(H)$, is

$$M(H) = M_S \left( \cot h \frac{\mu_0 H M_p V_{p1}}{K_0 T} - \frac{K_0 T}{\mu_0 H M_p V_{p1}} \right), \tag{12}$$

where $M_p$ is the magnetization of the solid phase of the MR fluid, $M_S$ is the saturation magnetic flux density of the MR fluid, $V_{p1}$ is the volume of each solid particle, and T is the absolute temperature of the MR fluid.

According to the Maxwell equation of MR fluid [24], $K(H)$ can be acquired as follows:

$$K(H) = \frac{1}{f} + \frac{1}{Z_r} \mu_0 H_0 M(H). \tag{13}$$

By substituting (13) into (11), the additional friction torque generated by the MR fluid can be obtained as

$$\Delta T = \frac{\frac{1}{f} + \frac{1}{Z_r} \mu_0 H_0 M(H)}{V H_0 M(H) K_0}. \tag{14}$$

Then, the frictional power consumption, $P$, of the MR fluid seal of the main shaft of the hydraulic turbine is

$$P = \omega_0 \times \Delta T \tag{15}$$

By substituting the specific-heat capacity and mass of the MR fluid in the MR fluid sealing gap of the turbine main shaft into (15), the frictional heat temperature value can be obtained.

## 3. Introduction to Models and Boundary Conditions

### 3.1. Model Introduction

The research object of this study is a tubular turbine generator set. The diameter of the main shaft of the turbine is 500 mm, the total length of the main shaft is 4000 mm, and the length of the sealing section is 1000 mm. Considering the design and processing of the actual test device, the test device is 1/10 of the real machine model. The test device is shown in Figure 3.

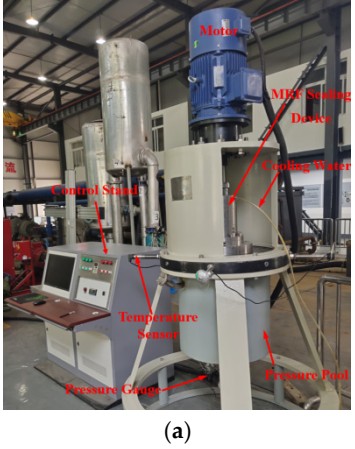 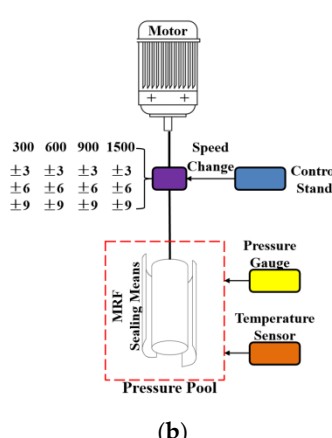

(a)    (b)

**Figure 3.** Experimental bench of MR fluid sealing device for main shaft of hydraulic turbine: (**a**) photograph; (**b**) diagram.

### 3.2. Numerical Calculation Model Introduction

The numerical calculation model adopts a scale model such as an MR fluid sealing test device. The calculation model is shown in Figure 4, and the device structure size is shown in Table 1.

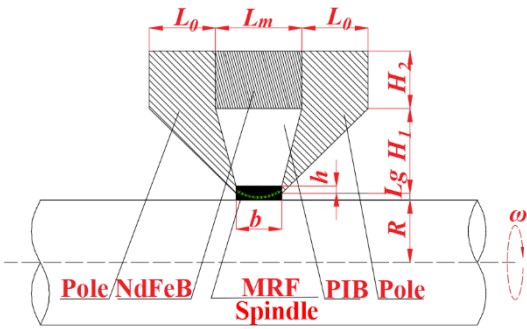

**Figure 4.** Partial view of MR fluid sealing device.

**Table 1.** Dimensions of MR Fluid Sealing Device (unit: mm).

| Axial Length of Pole/$L_0$ | Axial Length of Permanent Magnet/$L_m$ | Axial Length of Gap/$b$ | Spindle Radius/$R$ | Radial Length of Gap/$Lg$ | Pole Wedge Width/$h$ | PIB Radial Width/$H_1$ | Radial Width of Permanent Magnet/$H_2$ |
|---|---|---|---|---|---|---|---|
| 20 | 20 | 10 | 50 | 0.5 | 2 | 22 | 10 |

### 3.3. Setting of Boundary Conditions

First, through experiments, the viscosity–temperature curve and magnetic temperature curve of the MR fluid at different temperatures and external magnetic field strengths are measured; then, a function is written into the software through the UDF custom function in ANSYS; finally, an ANSYS co-simulation of the MR fluid is performed with the Maxwell and Fluent software.

The magnetic field setting conditions of the MR fluid sealing device are as follows. The permanent magnet adopts axial magnetization from left to right, the magnetic field magnetization relationship is the residual magnetic flux density, and the residual magnetic flux density is 1.21 T. Furthermore, the magnetic pole and spindle magnetic field magnetization relationship is the BH curve, which comes with the system. The rubber is an insulating material, and the magnetic field magnetization relationship is relative permeability, which is set to 1. The magnetic field magnetization relationship of the MR

fluid is relative permeability, and the specific value is the magnetic temperature curve determined through the experiment.

The setting conditions of the temperature field of the MR fluid sealing device are shown in Table 2. The viscosity of the MR fluid is determined by the test, and the specific value is the viscosity–temperature curve determined using the test.

**Table 2.** Thermodynamic Parameters of Various Materials in Temperature Field Simulation.

| Density of MRF/$\rho_m$(kg·m$^{-3}$) | Thermal conductivity of MRF/$\lambda_m$(W·(m·K)$^{-1}$) | Density of 45$^{\#}$steel/$\rho_s$(kg·m$^{-3}$) | Constant pressure heat capacity of 45$^{\#}$steel $C_s$(J·(kg·K)$^{-1}$) | Density of NdFeB/$\rho_N$(kg·m$^{-3}$) |
|---|---|---|---|---|
| 1200 | 0.2 | 7800 | 450 | 7550 |
| **Thermal conductivity of 45$^{\#}$steel/$\lambda_s$(W·(m·K)$^{-1}$)** | **Thermal conductivity of NdFeB/$\lambda_N$(W·(m·K)$^{-1}$)** | **Constant pressure heat capacity of NdFeB $C_N$(J·(kg·K)$^{-1}$)** | **Thermal conductivity of PIB$\lambda_P$(W·(m·K)$^{-1}$)** | **Density of PIB/$\rho_P$(kg·m$^{-3}$)** |
| 50 | 9 | 440 | 0.15 | 815 |

## 4. Test Results and Discussion

### 4.1. Sealing-Pressure Test Results and Discussion

The pressure values of the measuring points were tested under the conditions of 300, 600, 900, and 1500 rpm. The test results are shown in Figure 5. The figure also shows that, as the speed gradually increases from 300 to 1500 rpm, the pressure of the MR fluid sealing device for the main shaft of the turbine gradually decreases from 0.12 MPa to zero. By observing the sealing pressure of different variable amplitudes under the same speed condition, the speed change under the low-speed condition has no effect on the sealing pressure. For the high-speed condition, the speed change amplitude has a significant effect on the sealing pressure. Simultaneously, under the same rotational-speed conditions, the magnitude of the change in rotational speed is negatively correlated with the magnitude of the sealing pressure. When the magnitude is larger, the sealing pressure is smaller.

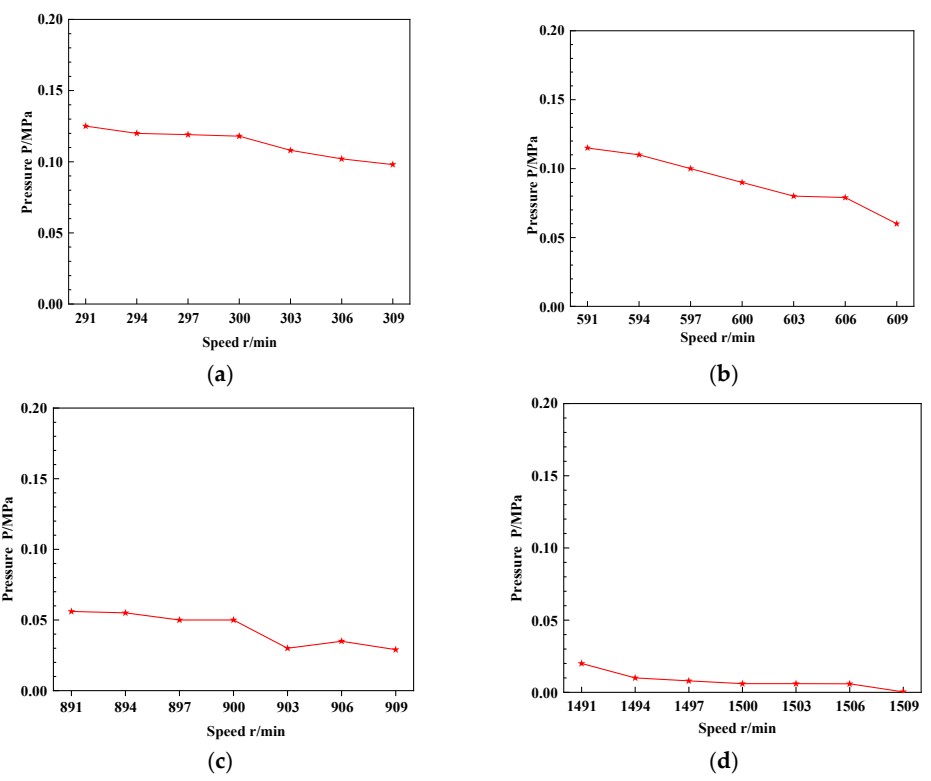

**Figure 5.** Pressure curve of the MR fluid seal under different speed conditions: (**a**) 300 rpm; (**b**) 600 rpm; (**c**) 900 rpm; and (**d**) 1500 rpm.

This phenomenon occurs for the following reason. Based on (5), it can be seen that, as the speed increases, when the centrifugal force term is greater, the pressure of the MR fluid sealing device will gradually decrease. For the same rotational speed operating condition, the rotational speed changes suddenly. Under the influence of inertial force, the MR fluid in the fluid seal device deflects, causing the magnetic moment of the solid particles in the MR fluid to change, thereby affecting the saturation magnetic induction intensity of the MR fluid, ultimately affecting the sealing pressure of the device.

### 4.2. Results and Discussion of the Frictional Heat Experiment

The temperature change in the MR fluid sealing device under different speed conditions is shown in Figure 6. The figure also shows that, as the rotation speed increases, the temperature in the sealing gap gradually increases from 46 to 76 °C. For low-speed operating conditions, when the device speed changes suddenly, the temperature remains unchanged. However, for high-speed operating conditions, after the device's rotational speed changes suddenly, its temperature suddenly changes, then stabilizes.

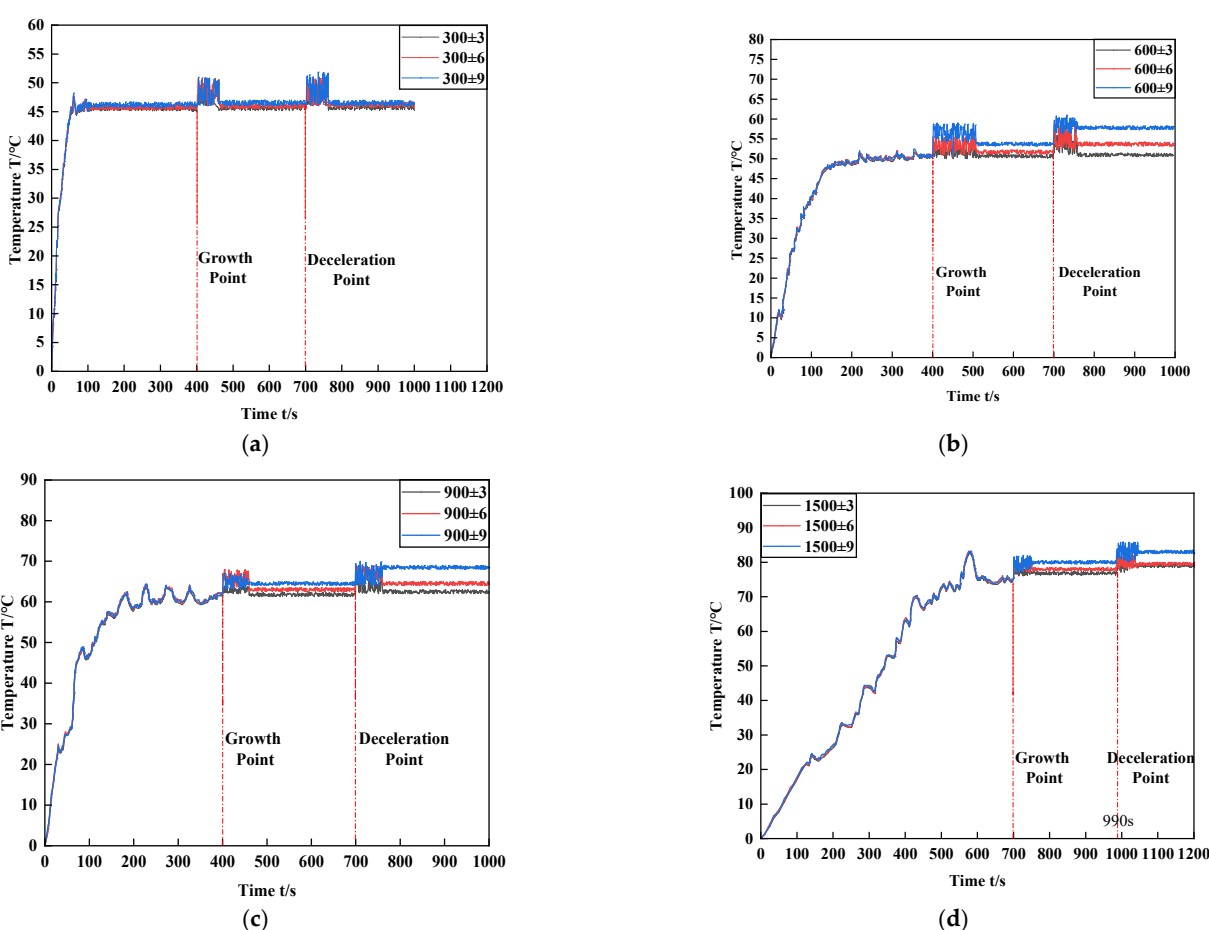

**Figure 6.** Temperature change in the MR fluid seal under different speed conditions: (**a**) 300 rpm; (**b**) 600 rpm; (**c**) 900 rpm; and (**d**) 1500 rpm.

This is due to the fact that, as the rotation speed increases, the friction between the MR fluid in the sealing gap, rubber, and magnetic poles intensifies, resulting in an increase in frictional heat. For abrupt conditions, the MR fluid in the sealing gap suddenly increases (decelerates), and the magnetic moment of the magnetic particles in the MR fluid changes under the action of inertial force, causing the MR fluid to malfunction, balancing the curl, generating heat, and causing the overall temperature to rise. As the main shaft speed of the turbine stabilizes, the magnetic moment of the magnetic particles in the MR fluid becomes consistent; however, the thermal effect caused by the unbalanced curl accumulates

in the sealed gap, causing the temperature to rise. Additionally, when the amplitude of the variable working condition is larger, the recovery process of the MR fluid is slower, the thermal effect caused by the change in the magnetic moment of the MR fluid is greater, and the temperature rise of the device is more evident.

## 5. Numerical Calculation and Analysis

### 5.1. Model Validation

According to the test results in the previous section, this study selects the temperature test value of the MR fluid and the numerical results under the 300-rpm speed mutation condition for comparison and verification. The verification results are shown in Figure 7. This figure shows that the time of the numerical calculation curve is delayed by 100 s compared to the test curve; nevertheless, the overall trend of the numerical calculation is consistent with the experimental trend, and the temperature value after stabilization is the same. However, for the numerical analysis results in this section, only the steady-state temperature and magnetic field of the MR fluid sealing device are discussed, and the change process is not discussed. The temperature after the calculation curve and test curve are stable and fundamentally the same; hence, the numerical calculation results are consistent with the experiment results.

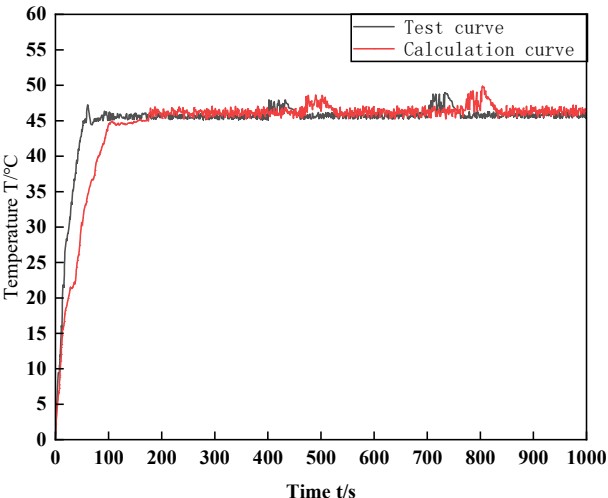

**Figure 7.** Comparison of the test and calculated temperatures of the MR fluid under 300 rpm.

### 5.2. Results and Analysis of Friction Heat

Figure 8 shows the temperature cloud diagram of the MR fluid sealing device under the condition of an external magnetic field, H = 0, at different speeds. As shown in Figure 8, as the rotation speed increases, the temperature of the MR fluid sealing device gradually increases, and the high-temperature area gradually increases. Simultaneously, by observing the same rotational speed, the maximum temperature of the MR fluid sealing device is noted in the sealing gap, which is approximately 55 °C; it then diffuses to the outside to realize heat exchange with the outside world.

Figure 9 shows the temperature cloud of the MR fluid sealing device under the condition of an external magnetic field, H ≠ 0, at different speeds. As shown in Figure 9, as the rotation speed increases, the temperature of the MR fluid sealing device gradually increases, and the high-temperature area continues to increase. Simultaneously, by observing the same rotational speed, the maximum temperature of the MR fluid sealing device is noted in the sealing gap, which is approximately 75 °C; it then diffuses to the outside to achieve heat exchange with the outside world.

Comparing and observing the cloud diagram of the MR fluid sealing device at different speed under the conditions of H = 0 and H ≠ 0 shows that the temperature rise of the MR

fluid sealing device under the action of an external magnetic field is significantly higher than that of the device without the action of an external magnetic field.

The main reason for this is that the main shaft of the hydraulic turbine moves in a circle, the MR fluid in the gap moves with the main shaft under the action of friction, and the magnetic poles and rubber are stationary parts. Hence, the MR fluid and magnetic poles in the gap move together. Moreover, the rubber exhibits a strong friction effect. According to (14), after the MR fluid is subjected to an external magnetic force, it produces an unbalanced curl. Therefore, for an MR fluid with magnetic force under the dual action of friction and unbalanced curl, the frictional power consumption increases exponentially, resulting in a significant increase in the temperature of the MR fluid at the gap. Additionally, an increase in the temperature of the sealing device causes the magnetic properties of the magnetic poles, permanent magnets, spindle, and MR fluid to decrease, reducing the pressure resistance of the seal. This observation corresponds to the results of the magnetic-field analysis.

### 5.3. Magnetic Field Results and Analysis

Figure 10 shows the calculation results of the magnetic field of the MR fluid sealing device under different speed conditions. As shown in this figure, with an increase in the rotation speed of the main shaft of the turbine, the magnetic field of the MR fluid sealing device is generally produced by the N level of the permanent magnet, followed by the magnetic pole → MR fluid → main shaft → MR fluid → magnetic pole; then, it returns to the S level. However, the high magnetic-field area of the MR fluid sealing device under low-speed operating conditions is significantly larger than that under high-speed operating conditions; with a further increase in the rotational speed, the magnetic field area remains unchanged and tends to be stable.

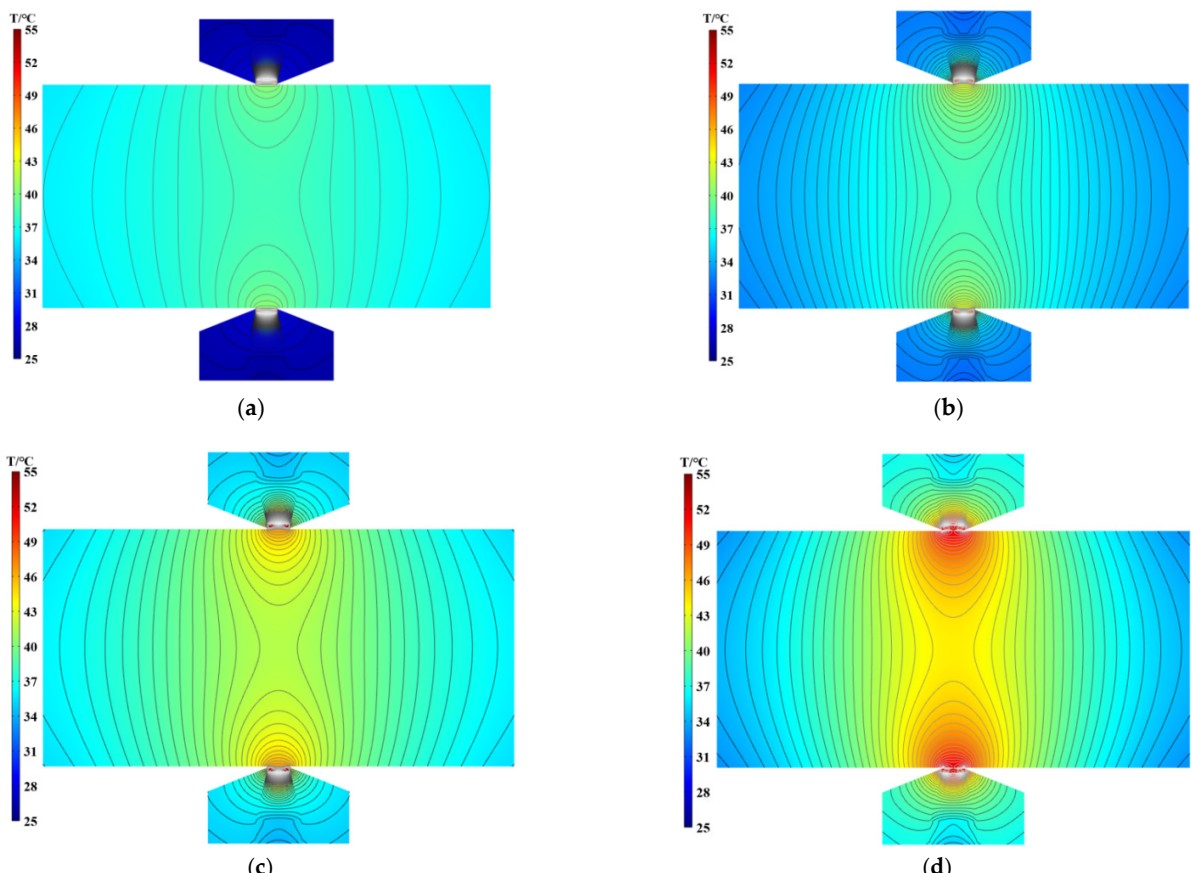

**Figure 8.** Temperature cloud of the MR fluid seal at different speeds with H = 0: (**a**) 300 rpm; (**b**) 600 rpm; (**c**) 900 rpm; and (**d**) 1500 rpm.

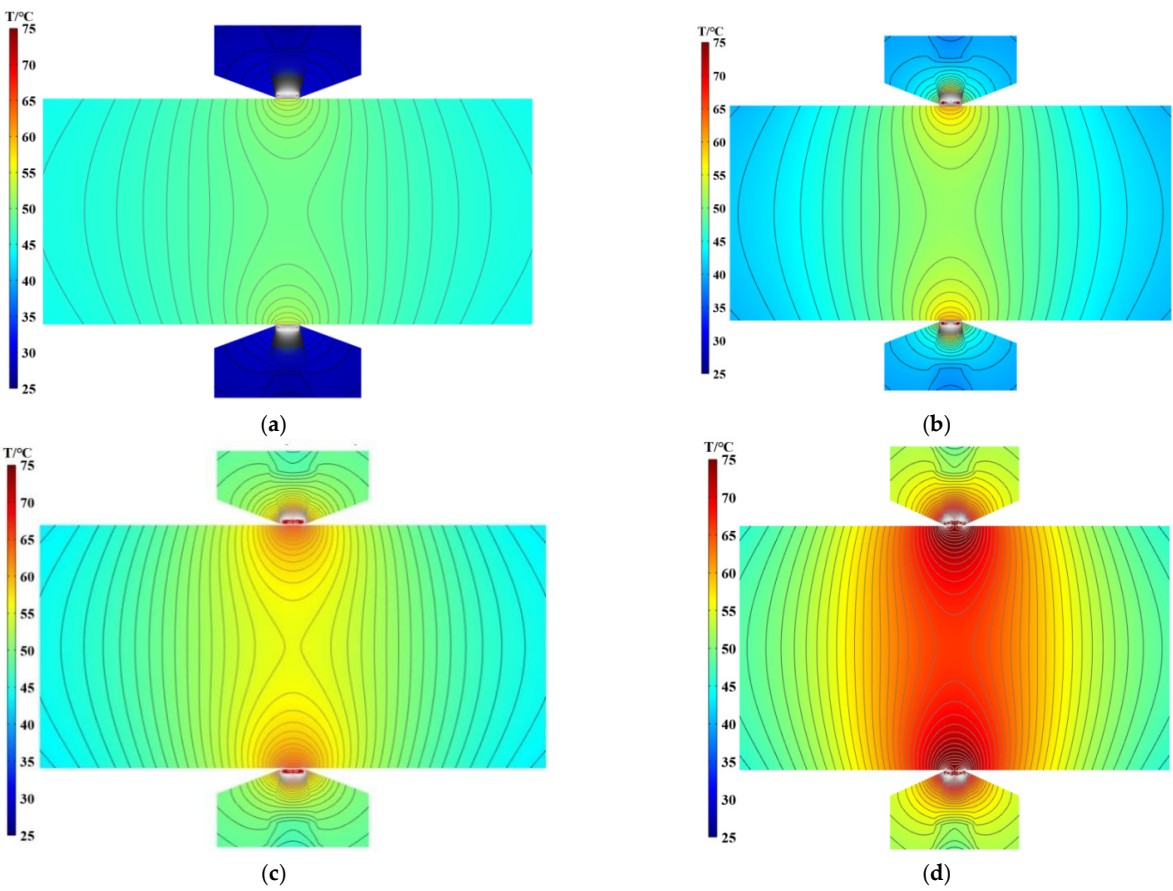

**Figure 9.** Temperature cloud of the MR fluid seal under different speed conditions with H ≠ 0: (**a**) 300 rpm; (**b**) 600 rpm; (**c**) 900 rpm; and (**d**) 1500 rpm.

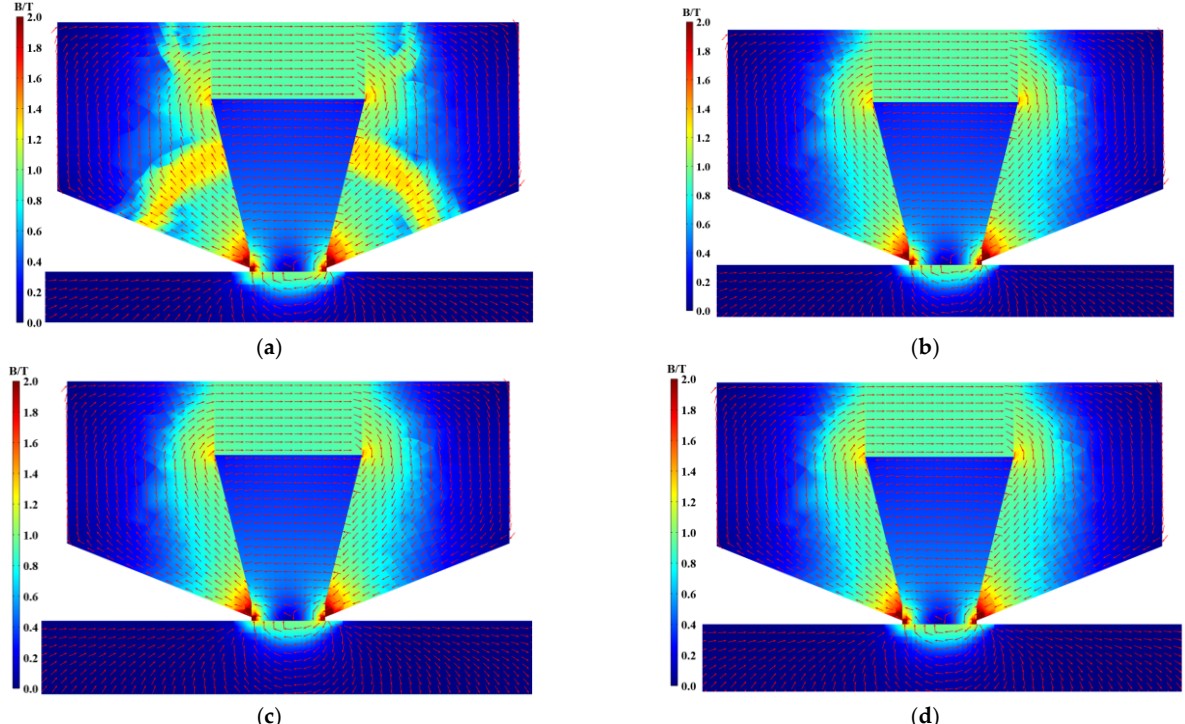

**Figure 10.** Magnetic-field cloud of the MR fluid seals at different speeds: (**a**) 300 rpm; (**b**) 600 rpm; (**c**) 900 rpm; and (**d**) 1500 rpm.

Figure 10 shows that the magnetic field results of high-speed operating conditions are approximately the same. This is due to the fact that, with a continuous increase in speed, the temperature of the MR fluid continues to rise, and the relative permeability of the MR fluid gradually decreases from 1.05 to 1. However, for the magnetic poles, spindles, rubber, and permanent magnets, a small range of temperature change has little effect on their magnetic properties. Therefore, for the entire MR fluid sealing device, the magnetic field area is unchanged under high-speed working conditions. Therefore, for further research on the magnetic field, the field results of 300 and 600 rpm operating conditions were selected for this study. According to the literature [25], it is known that the side closest to the main shaft easily causes leaks. Therefore, in this study, a section line close to the main shaft (0.05 mm) with an axial length of 10 mm is examined. The value of the magnetic field on the section line is shown in Figure 11.

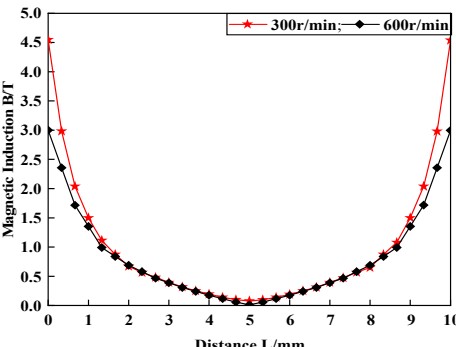

**Figure 11.** Magnetic field of the sealing gap under 300 and 600 rpm speed conditions.

Figure 11 shows that the magnetic field at the sealing gap at 300 and 600 rpm changes in an upward parabola, the magnetic induction intensity in the middle of the gap is close to zero, and both ends of the gap are the maximum value of the magnetic field, at 4.5 and 3.0 T. The magnetic field at both ends of the 300 rpm scenario is greater than that of the 600 rpm scenario.

Finally, according to the results of the magnetic-field analysis, combined with (5), the sealing pressure at 300 rpm is noticeably greater than that at 600 rpm. The reason for this is that, as the spindle speed increases, the friction power consumption of the MR fluid at the gap increases. The temperatures of the magnetic poles, permanent magnets, and spindle materials also increase, and the magnetic performance decreases, resulting in a decrease in the sealing pressure. When the rotation speed increases to a larger value (600 rpm), the magnetic properties of the magnetic poles, permanent magnets, and spindles drop to a certain threshold, and appear unchanged, as shown in Figure 10.

## 6. Conclusions

This study first derives the MR fluid seal pressure and unbalanced curl equations of the hydroturbine main shaft, then uses experiments to analyze the seal pressure and friction heat under different rotational-speed mutation conditions. The friction heat test results and numerical calculation results are analyzed. After verification, the temperature field and magnetic field distribution of the MRl fluid sealing device of the main shaft of the hydraulic turbine are obtained using numerical calculation. The results obtained are as follows.

(1) For the MR fluid sealing device of the hydraulic turbine main shaft, when the spindle speed changes randomly, it causes the magnetic moment of the magnetic particles in the MR fluid to change, resulting in unbalanced curl friction power consumption and a rise in device temperature, which leads to sealing failure;

(2) For turbines having random variable speed conditions, the spindle speed and variation amplitude of the turbine are directly proportional to frictional heat, which reduces the sealing ability of the device, leading to sealing failure;

(3) For medium- and high-speed turbines, if an MR fluid sealing device is used, the necessary cooling for the device can be improved, and the stability of the operating conditions can be increased to reduce the frictional heat and increase the sealing life of the device.

As for the research on the MR fluid sealing, most existing studies are conducted on the stability of the interface between the sealing medium and magnetic fluid. However, MR fluids, which have dual characteristics of fluids and magnetic materials, can be investigated from the perspective of both characteristics. This study is conducted through the influencing factors (temperature) of the magnetic properties of magnetic materials; hence, in the future, it can be studied based on the influencing factors of MR fluids. In addition, for the main body of the study, the main shaft of the turbine is the only application object. In future studies, this research method can be applied to all sealing industries.

**Author Contributions:** Z.-G.L. proposed the ideas and method of the research, and financial support; J.C. wrote the original paper and contributed to the review and editing; Y.X. verified the test results; W.-X.L. supervised and managed the test; and X.-R.L. was responsible for the processing and design of the test device. All authors have read and agreed to the published version of the manuscript.

**Funding:** This research was funded by the National Natural Science Foundation of China (Grant No. 52079118), Sichuan Provincial Department of Science and Technology International Cooperation Project (Grant No. 2020YFH0135), and Sichuan Science and Technology Innovation and Entrepreneurship Project (Grant No. 2020043).

**Institutional Review Board Statement:** Not applicable.

**Informed Consent Statement:** Not applicable.

**Data Availability Statement:** The study did not report any data.

**Acknowledgments:** The authors would like to thank the National Natural Science Foundation of China (Grant No. 52079118), Sichuan Provincial Department of Science and Technology International Cooperation Project (Grant No. 2020YFH0135), and Sichuan Science and Technology Innovation and Entrepreneurship Project (Grant No. 2020043).

**Conflicts of Interest:** The authors declare no conflict of interest.

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
