# Peer review of "Study on the Unbalanced Curl Seal Failure of the Magnetorheological Fluid Sealing Device of the Hydraulic Turbine Main Shaft under Different Speed Abrupt Conditions"

_processes, doi:10.3390/pr9071171_

Round 1

Reviewer 1 Report

 I think the paper can be published after the following observations are corrected:

  • Pay attention to the notations used to explain the equations. For example, when explaining equation 1 you used ρm with no m subscript, ηH you did not put H subscript.
  • In Equation 1 you did not explain what p and miu mean (pressure and permeability). All the terms need to be explained even for more uninformed readers.
  • Be clearer in the expression in lines 112 and 113. Eventually, add an IS and a THEN: “when the rotation radius IS r = R THEN V = R omega”
  • Also, the expression in lines 119, 120 “by combining (4) and the yield stress formula… the MR fluid can be obtained” is not correct. You do not get the MR fluid but you get the equation of the pressure variation in the fluid.
  • Use subscripts where necessary in explaining equation 5. Use subscript indices where necessary in explaining all equations.
  • Line 191 - "rotation of the runner in the runner" - I don't think the expression is ok.

I consider that the paper approaches a very narrow technical field: the problems that appear as a result of the thermal effect in the sealing liquid inside the hydroelectric turbines - effect seems to be caused by the magnetic properties of the liquid. This is why I think it would be extremely suggestive to introduce future research directions (starting from the ideas in the paper). So enter a Future Trends at the end of the Conclusions section that would frame the paper in a more general framework and that could interest more readers.

Reviewer 2 Report

In the manuscript MR fluid sealing device is investigated numerically and through experiments.

The modelling parameters and modelling procedure, as well as well model parameters are not described in the manuscript.

Can authors please explain why the magnetic field area remains basically unchanged at low-speed and high-speed operating conditions in Figure 5.

The structure of the manuscript is to be changed. The experimental setup should be presented not in results and discussion. In results and discussion only results are to be discussed.

I could not find verification of the modelling results through the experiments in the manuscript. The section with the experiments is not connected with the modelling section. Through the experiments the modeling results are to be confirmed.

Round 2

Reviewer 2 Report

The manuscript can be accepted in present form.